# Estimating future wildfire burnt area over Greece using the JULES-INFERNO model

Anastasios Rovithakis[1,2], Eleanor Burke[4], Chantelle Burton[4], Matthew Kasoar[2,3], Manolis G. Grillakis[1,2], Konstantinos D. Seiradakis[1,2], Apostolos Voulgarakis[1,2,3]

5   [1]School of Chemical and Environmental Engineering, Technical University of Crete, Chania, Greece.
[2]Leverhulme Centre for Wildfires, Environment and Society, Imperial College London, London, UK.
[3]Department of Physics, Imperial College London, London, UK.
[4]Met Office Hadley Centre, London, UK.

*Correspondence to*: Anastasios Rovithakis (arovithakis@tuc.gr)

10   **Abstract.** Our previous studies have shown that fire weather conditions in the Mediterranean and specifically over Greece are expected to become more severe with climate change, impling potential increases in burnt area. Here, we employ the Joint UK Land Environment Simulator (JULES) coupled with the INFERNO fire model driven by future climate projections from the UKESM1 model to investigate the repercussions of climate change and future vegetation changes on burnt area over Greece. We validate modelled burnt area against the satellite-derived GFED5 dataset, and find the model's performance to be good, especially for the more fire-prone parts of the country in the south Greece. For future simulations, we use future climate data following three Shared Socioeconomic Pathways (SSPs), consisting of an optimistic climate change scenario where fossil fuel emissions peak and decline beyond 2020 (SSP126), a middle-of-the-road scenario (SSP370), and a pessimistic scenario where emissions continue to rise throughout the century (SSP8.5). Our results show increased burnt area in the future compared to the present-day period in response to overall hotter and drier climatological conditions. We use an additional JULES-INFERNO simulation in which dynamic vegetation was activated, and find that it features smaller future burned area increases compared to our simulation with static present-day vegetation. For this dynamically changing vegetation simulation the greatest burnt area increases are found for southern Greece, due to higher future availability of flammable and heat-resistant needleleaf trees and the smallest decreases in agricultural areas of northern Greece due to a reduction in the aforementioned tree category.

**1 Introduction**

For some ecosystems, wildfires can have a positive impact serving as a catalyst for plant life regeneration (Lelieveld et al., 2002; Littell et al., 2010). However, for the vulnerable ecosystems, wildfires can have devastating effects (Andela et al., 2018). Furthermore, wildfires can cause negative consequences when it comes to the atmospheric environment (Forkel *et al.*, 2019; Voulgarakis and Field, 2015), human health (Chuvieco *et al.*, 2018; Reid *et al.*, 2016) and the economy (Nielsen-Pincus, Moseley, and Gebert 2014). In Mediterranean type of environments, future temperature increases and precipitation decreases

have been shown to lead to increased future fire danger (Turco et al., 2018; Batllori1 et al., 2013). Specifically, for the area of Greece, which features a typical Mediterranean climate and Mediterranean ecosystems, our previous work has estimated that future climate change (e.g. temperature increases of 2 – 5 °C - Zittis et al., 2019), could lead to increases in future fire season length of up to a month for some areas (Rovithakis et al., 2022). The combination of increased drought frequency and intensity, coupled with rising temperatures, creates conditions highly conducive to larger and more severe fire events (Resco de Dios et al., 2022). Studying and understanding how future burnt area might evolve is important for providing insight into the potential future effects of wildfires. Apart from the better known impacts on infrastructure, ecosystems, air quality and health, fires can also affect local temperatures due to radiative forcing resulting from their emissions (M. G. Tosca, D. J. Diner, M.J.Garay 2014; Tosca, Randerson, and Zender 2013; Jiang et al. 2020). Moreover, wildfires' ability to affect soil structure can cause enhancements in runoff thus increasing the future likelihood of flash flooding and further infrastructure destruction (Neary et al. 2012; Langmann et al. 2009; Pfister, Wiedinmyer, and Emmons 2008; Grillakis et al,. 2024). The Mediterranean basin and specifically Greece is one of the hot-spots of global climate change (Lelieveld et al., 2002; Lelieveld et al., 2001), while at the same time being in the crossroads of many different atmospheric pollution types such as fine anthropogenic aerosols and ozone precursors from Europe, desert dust from North Africa and the Middle East, and maritime aerosols from the Mediterranean Sea and the Atlantic Ocean (Kalivitis et al., 2007).

Changes in landscape fragmentation and connectivity patterns are complex and vary depending on the specific area and drivers. While urban expansion and sprawl can contribute to land fragmentation (De Montis et al., 2017), other processes, particularly the abandonment of formerly utilized land, can lead to a decrease in fragmentation and an increase in landscape homogeneity as diverse land use types converge towards more uniform vegetation covers, such as dense shrublands or forests (Hill et al., 2008). On the other hand, fire suppression practices in the region can contribute to fuel accumulation over longer periods by preventing more frequent and less intense fires, leading to fuel build-up on abandoned agricultural lands (Salis et al., 2022). Future land use scenarios project varying changes in total agricultural land globally by 2100, including potential increases or decreases, depending on the specific pathway (Hurtt et al., 2011). Technological advances such as irrigation have historically led to the expansion of agriculture in some areas (Hill et al., 2008). In these human-influenced areas, fire can increase the landscape's homogeneity thus increasing the extent of future burnt area (Loepfe et al., 2010).

Other studies on the effects of land use/land cover changes on wildfires on a global level found decreasing fire emissions in response to harvested land cover change and increases in response to future climate change (Kloster et al., 2012). In the United States, in regions where urbanized areas replace forests and grasslands, it is expected that there will be increased surface temperature and vapour pressure deficit, along with reduced precipitation compared to the current land use/land cover pattern. Conversely, in regions where croplands replace forests, the opposite tendency is observed. These alterations in local and regional atmospheric conditions can result in extended fire season and more frequent and intense wildfires (Zhong et al., 2021; Bryant and Westerling, 2014). Mediterranean ecosystems, have experienced increased fire occurrence in response to changes in the agricultural/forest interface and urban/forest interface (Gallardo et al., 2016). The current study aims to predict potential future changes of burned area over Greece under different climate change scenarios using a fire-enabled dynamic global

vegetation model (DGVM), and the implications of using varying versus static land use for these predictions. This comparative approach allows for disentangling the direct climate-driven impacts on fire weather and flammability from those mediated by climate-induced shifts in vegetation composition and fuel availability. Such a distinction is insightful, as climate change can alter vegetation distributions, thereby changing fuel loads and types, which changes fire behaviour in ways that static vegetation approaches cannot capture (Rogers et al., 2020). Past studies (mainlyKarali *et al.*, 2014, Di Giuseppe *et al.*, 2020 and our recent

work by Rovithakis et al., 2022) have relied on the use of fire danger indices (specifically FWI) for predicting future fire danger changes. The novelty of this work is that it utilizes a fire-enabled DGVM, so it can account for vegetation changes, along with the climate-driven fire changes. Additionally, the model can predict burned area changes, which is not a capability offered by fire danger indices used in past studies.

## 2 Methodology

### 2.1 Overview of JULES-INFERNO

For this study, the JULES-INFERNO modelling system (Mangeon *et al.*, 2016; Burton *et al.*, 2019) was utilized to perform future burnt area estimates. The Joint UK Land Environment Simulator (JULES) is an advanced land surface model (LSM) developed to simulate the dynamics of terrestrial hydrology, vegetation, carbon storage, and the surface exchange of water, energy, and carbon, as outlined by Clark et al. (2011). Moreover, JULES integrates the INteractive Fire and Emission

algoRithm for Natural environments (INFERNO), which estimates fuel flammability based on a simplified fire count model influenced by monthly average temperature, relative humidity, fuel load, soil moisture and precipitation. That in conjunction with fire ignitions based on human population density and lightning strikes helps the model diagnose burnt area. Other weather variables such as wind speed are not as important when simulating the collective effect of multiple fires on large scales (grid-scale level for regional studies in this case), as the core of fire occurrence probability is largely linked to moisture conditions

and ignition sources (Pechony and Shindell 2009). Observational evidence has shown fire size to be strongly linked to land cover (Chuvieco, Giglio, and Justice 2008; (Giglio, Randerson, and Van Der Werf 2013) so, specific average burnt area was assigned to each Plant Functional Type in order to ensure the model simulates larger fires in grasslands and shrublands compared to forests (Mangeon et al., 2016). This algorithm also considers human population density and lightning as sources of ignition. In INFERNO, upper soil moisture reflects the residual effects of past precipitation, which contrasts with immediate

rainfall that acts as a quick fire suppressant. Traditional measures of vegetation density are replaced by a fuel load index reliant on leaf carbon and decomposable plant matter, or litter, including surface and canopy fuel which are estimated internally by JULES. INFERNO's ignition processes include variables for both anthropogenic and natural causes, specifically lightning, as detailed by Mangeon et al., (2016).

## 2.2 Plant functional types and fire response traits

Within JULES, the TRIFFID (Top-down Representation of Interactive Foliage and Flora Including Dynamics) DGVM is integral for simulating the carbon cycle and the distribution of various plant functional types (PFTs) and their interactions, including growth, competition, and mortality, to assess how vegetation influences fire dynamics and vice versa (Burton et al., 2019). Plant functional types (PFTs) are categories used in land surface and dynamic global vegetation models, such as TRIFFID, to represent vegetation types (Clark et al., 2011). Instead of representing individual plant species, PFTs group plants

based on shared functional and structural characteristics. This approach allows models to represent vegetation heterogeneity within grid boxes in a computationally efficient way (Mathison et al., 2023). The main characteristics to define PFTs are the photosynthetic pathway to categorize them as C3 or C4; the phenology based on which they are classified as deciduous or evergreen; the life form based on which we have trees (broadleaf, needleleaf), shrubs, and grasses; the climate zone adaptation such as temperate or tropical for broadleaf evergreen trees; and land use/management based on which we have managed (crops,

pasture) or natural. Managed PFTs such as crops might be treated differently, for instance, assumed not to be nitrogen limited with litter removed as a simple representation of harvest (Burton et al., 2019).

When fires cause vegetation mortality, TRIFFID tracks the impact on biomass and carbon stocks, while modelling post-fire regrowth that alters future fuel availability and potential fire behaviour (Mathison et al., 2023). TRIFFID also accounts for fire-climate interactions by incorporating factors like temperature, humidity, and soil moisture, which influence vegetation

flammability and fire spread across different PFTs under varying climate conditions (Burton et al., 2019). In JULES-INFERNO simulations with static vegetation cover, prescribed vegetation types typically represent a predefined distribution of plant functional types (PFTs) across the landscape. This setup contrasts with dynamic vegetation models like TRIFFID, where vegetation can change in response to climate and fire feedbacks. In a static vegetation setup, the vegetation cover is assigned based on observed or assumed conditions and does not change over time.

Each PFT is associated with unique fire response traits (e.g., flammability, fuel load) (Mangeon et al., 2016; Mathison et al., 2023). INFERNO calculates the rate of burning, represented as the fraction of gridbox burned per second ($s^{-1}$), by multiplying the flammability of vegetation by the ignition rate (ignitions per kilometer per second) and the average burned area per fire. This calculation yields a fractional burning rate across the gridbox rather than tracking individual fires. Ignitions are treated as a continuous rate per unit distance and time, meaning that ignition events are not discrete but constant within each time step.

This is important in order to determine how the increase in fire occurrence, burnt area, and fire season length are driven by climate-induced changes in vegetation flammability as opposed to climate-induced changes in vegetation dynamics. Consequently, this ignition rate is scaled by vegetation flammability to determine the frequency of fires. The burning rate across the gridbox is then calculated by multiplying the fire initiation rate by the average area burned per fire, resulting in a burning fraction per second across the grid area (Mangeon et al., 2016).

Output files from JULES-INFERNO include information for various classes representing distinct land cover types and vegetation functional types. For instance, variable 'evgndltr' denotes the total evergreen needle leaf trees along with the litter

(fallen leaves and needles) associated with them. The class 'evgbdltr' indicates the evergreen broadleaf trees including their associated litter. 'C4grass' refers to grasses utilizing the C4 photosynthetic pathway, typically found in warmer climates and more efficient in photosynthesis under high temperatures and intense light than C3 grasses. Conversely, 'C3grass' represents grasses using the C3 photosynthetic pathway, which are more prevalent in cooler, wetter environments and less efficient under high temperatures and light conditions compared to C4 grasses. 'C4crop' signifies crops that follow the C4 photosynthetic pathway, resembling C4 grasses in their efficiency under high-temperature and light conditions. 'C3crop' pertains to crops employing the C3 photosynthetic pathway, common in cooler, wetter environments, and less efficient under high temperatures and light compared to C4 crops. 'Dcdndltr' represents the total deciduous needleleaf trees including litter associated with these trees. 'Dcdcldbdltr' representing the total deciduous broadleaf trees including litter associated with these trees and the 'total' representing the total vegetation, including all the different types of vegetation classes (Best *et al.*, 2011; Clark *et al.*, 2011; Mangeon *et al.*, 2016).

### 2.3 Input data

Our simulation domain covers the entire globe, which then was cropped over the Greek domain to keep consistent Boundary Conditions, with a resolution of 0.5°. The simulations cover three 10-year periods, i.e. a reference period (1980-1990) and two future ones (2030-2040 and 2080-2090)  based on UKESM1-0-LL (which is the model from which future climate projections were taken) and on 3 future climate Shared Socioeconomic Pathways experiments (ssp126, ssp370 and ssp585). For these simulations, we utilized the advanced land surface model JULES (Joint UK Land Environment Simulator) configuration (u-cc669 at vn6.2) used in the Inter-Sectoral Impact Model Intercomparison Project (ISIMIP) 3b, in conjunction with the ISIMIP3b prescribed data at 0.5° (Stefan and Büchner 2024), ), but without the dynamic vegetation model TRIFFID in order to compare it with equivalent JULES-INFERNO simulations involving dynamic vegetation which have already been performed for ISIMIP3b. Comparison of our fixed-vegetation simulations with the ISIMIP simulations, will isolate the role of climate-driven flammability changes from the role of climate-driven dynamic vegetation changes (including the effects of future fire on vegetation) in driving future burned area changes in the area of Greece.

### 2.4 Model evaluation

For model evaluation purposes, we performed two additional simulations (one with and one without dynamic vegetation) for the recent observational period (2004-2019). These simulations are driven by observation-based reanalysis weather variables used in the ISIMIP3a modelling experiment. The simulated burnt area from these simulations was validated against the GFED5 observation-based burnt area dataset for the same period as the observations. This period was chosen in order to be in line with Y. Chen et al., (2023), as they found data from this period to be more consistent since both MODIS Terra and Aqua data were available. GFED5 is the newest version of the Global Fire Emissions Database, which uses the Terra and Aqua combined monthly burned area product (MCD64A1) as the base for calculating the 2001–2020 burned area, in combination with the fine-resolution burned area images from the program of Earth observation satellites such as Landsat or Sentinel-2 and MODIS

active fire data (Chen et al., 2023). Other studies have found that JULES-INFERNO shows realistic vegetation and burn

dynamics with strong spatial performance (Burton et al., 2019; Hantson et al., 2020) whilst capturing burnt-area trends (Mathison et al., 2023)

## 2.5 Calculation of burnt area and Fire Weather Index

To create the distribution of burnt area change for Figure 4, burnt area INFERNO outputs (monthly averaged from daily values) representing a reference period (1980-1990) and future periods (2030-2040 and 2080-2090) under different Shared

Socioeconomic Pathways (SSP126, SSP370, SSP585) were cropped over the Greek domain. The change in BA was calculated by subtracting the "past" burnt area from each future scenario's burnt area. These change values are then flattened into one-dimensional arrays and binned into predefined size categories (e.g., -1 to 0 km², 0 to 1 km² change). The final figure is a grouped bar chart where each group of bars corresponds to a BA change bin, and individual bars within the group show the frequency (number of instances/grid cells) of that change magnitude for the different SSP scenarios and future decades,

allowing comparison of how BA is projected to change under varying future conditions.

Figure 6(b) was made by calculating the spatial average yearly FWI values from the regional climate models (RCMs) used in our previous study (Rovithakis et al., 2022) for the fire season and that was done for each of the three 10-year time periods and for every RCP scenario separately. Since these are country wide averages, an increase by one EFFIS fire danger class is considered to be a drastic change. That way the potential severity in fire danger is better understood. A similar procedure were

followed for JULES-INFERNO modelled burnt area. The burnt area projections were generated using the JULES-INFERNO model driven by climate inputs corresponding to specific Shared Socioeconomic Pathways (SSPs) (i.e., SSP126, SSP370, SSP585) (O'Neill et al., 2016), which are coupled with Representative Concentration Pathways (RCPs) to define the overall climate forcing (Moss *et al.*, 2010; van Vuuren *et al.*, 2011). The Fire Weather Index (FWI) data is directly derived from climate model outputs forced by these RCPs (RCP2.6, RCP4.5, RCP8.5). Therefore, this comparison is valid as it examines

the relationship between a direct climate-driven fire danger metric (FWI) and a fire impact metric (BA) under consistent underlying climate change projections, allowing to assess the influence of similar climatic shifts on both fire weather and resultant burnt area  (Liu, Stanturf, and Goodrick 2010).

## 3 Results

### 3.1 Model performance

We first examined the performance of JULES-INFERNO in terms of simulating burnt area against GFED5 observations for years 2004-2019. For this comparison, atmospheric forcings from observation-based reanalysis GSWP3-W5E5 climate forcing part of the Inter-Sectoral Impact Model Intercomparison Project (ISIMIP3a) from the 'obsclim' climate experiment available in the ISIMIP Repository Lange et al. (2023) were utilised as input data to calculate burnt area in the simulation with static

vegetation, while archived burnt area output from ISIMIP3a (Burton et al., 2024) was obtained for the corresponding simulation with dynamically changing vegetation.

Even though there are disagreements of the JULES-INFERNO simulations when compared to the GFED5 observations, the former can still capture the general burnt area behaviour with an acceptable correlation for the majority of the areas in the Greek domain as seen in Figure 1 panels a and b, especially for the southern parts of the country. This level of performance,

capturing general trends while exhibiting regional variations in accuracy, is common in large-scale fire modelling studies when compared against satellite-derived products like GFED5 (Kloster *et al.*, 2012; Mangeon *et al.*, 2016). These two panels feature similar correlation patterns, with panel b representing the correlation with dynamically changing vegetation being slightly worse as also seen in Figure 1c, which shows the difference between results for dynamic minus static vegetation. That can be attributed to the additional degree of freedom in the dynamic vegetation simulation. The simulation with prescribed static

vegetation uses land cover observations to constrain the vegetation quantities, and since during the relatively short period of 2004-2019 the vegetation remains fairly steady, that simulation has slightly better correlation. The simulation with dynamic vegetation involves an additional uncertainty that leads to a departure from the true state, and the consequent bias. A limiting factor is that fire models (such as JULES-INFERNO) only rely on weather conditions and vegetation quantities to calculate burnt area, without any information on actual fire ignitions, a factor that is impossible to predict, due to its stochastic nature.

This, in turn, makes it impossible to predict e.g. the extremely high 2007 actual burnt area seen Figure 1 panel (d) to its full extent. This limitation regarding ignition and the precise replication of individual extreme events is a well-documented challenge (Mangeon et al., 2016). While models can simulate the potential for large fires under conducive weather, the exact timing and location of ignitions, and the subsequent fire spread influenced by fine-scale factors, are often beyond their scope without direct assimilation of ignition data. However, still, the tendency for increased burnt areas in years such as 2007 is

indeed captured by the model.

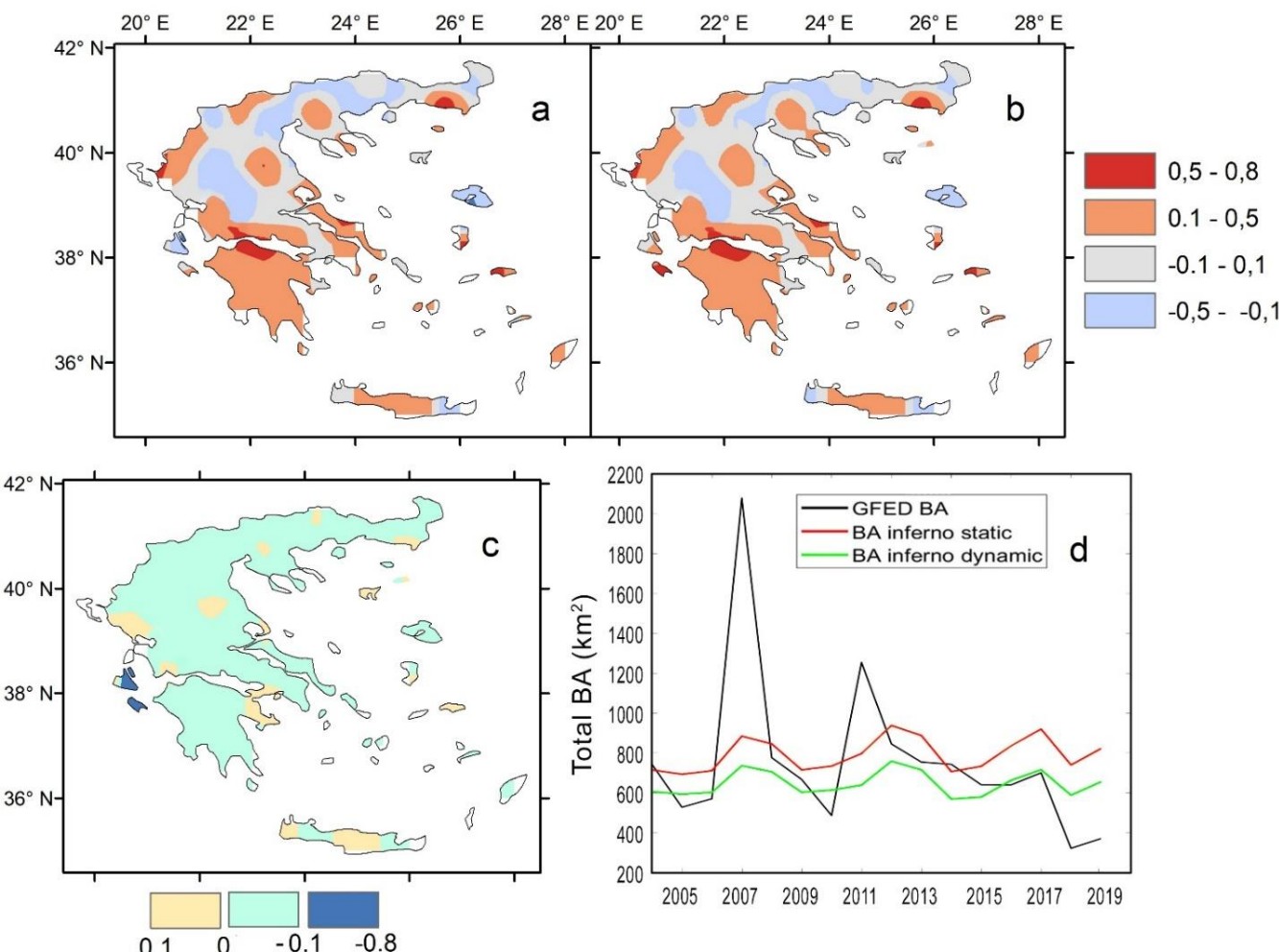

**Figure 1. Panels (a) and (b) show the temporal correlation (generated from monthly means) between GFED5 and JULES-INFERNO burnt area over Greece, for the simulation with static vegetation cover and the simulation with dynamically changing, respectively. Panel (c) shows the difference between the two correlations (panel (b) - (a)). Panel (d) shows comparison of annual average data for all years for the domain-wide spatial average, between the GFED observations and the simulated burnt areas with static and dynamically changing vegetation cover.**

## 3.2 Future changes and climatic drivers

Figures 2 and 3 (top panels) show the future changes of burnt area for the 2030-2040 period and for the 2080-2090 period, respectively. For the first experiment with static vegetation, the only factors that can influence burnt area are the climatological conditions. Future changes in these variables for the various scenarios are presented in the 2nd to 4th rows of panels in Figures 2 and 3.

Daily values of the decade 1980-1990 were subtracted from the daily values of 2030-2040 for figure 2 and similarly from
2080-2090 for figure 3 which showed that burnt area is projected to increase for all areas in Greece in the distant future when
compared to the reference period for all future scenarios except for the optimistic SSP126, which projects a small decrease for
some areas of up to 0.1 km2. A similar pattern as that for the SSP126 distant future panel emerges for all SSP scenarios for
the near future (panels a1-c1). Out of the panels (a1-c2) we see greater burnt area increases mainly in eastern continental
Greece. Those increases get more pronounced in the distant future for SSP370 and SSP585 (b2-c2) reaching up to 2.5 km2
additional burnt area compared to the reference period, which corresponds to an average 200% increase for those larger burnt
areas. The magnitude of these increases, particularly under higher emission scenarios like SSP585, show the potential for
substantial exacerbation of fire regimes, a result also found in studies projecting future fire danger across the Mediterranean
basin (Kloster *et al.*, 2012; Voulgarakis and Field, 2015).

When it comes to the drivers of burnt area change, we see that while temperature does generally increase in the country, panels
(d1-f2) do not follow the east west divide present in burnt area changes seen in the top panels of Figures 2 and 3. Instead, it
demonstrates a latitudinal gradient with higher temperature changes of up to 9 °C occurring in northern Greece, explaining the
somewhat boosted increases of burnt area in that area for all periods and scenarios. The pronounced warming, especially in
northern Greece, directly contributes to increased fuel aridity and flammability, a key mechanism by which climate change
influences fire activity. This strong temperature signal which is a primary driver of future fire risk is a recurrent theme in
regional climate impact assessments  (Zittis et al., 2019)

The relatively minor distant future burnt area decreases in SSP126 in parts of western Greece (panel a2) can be explained by
the domination of wetter conditions in the corresponding areas since this scenario is the most optimistic one and features more
precipitation (panels g2 and j2, respectively). This underlines the potential benefits of strong climate mitigation efforts in
moderating future wildfire risk.

However, overall, the changes in precipitation panels (j1-l1, j2-l2) are relatively small and thus appear to contribute little to
the burnt area changes. One example of that small contribution can be seen in panel l2 where under the most pessimistic
scenario the northeastern edge of Greece is projected to experience more precipitation, but also increases in burnt area seen in
panel c2.

Soil moisture is an additional metric that is being influenced by weather conditions. Precipitation directly adds water to the
soil, making it the most immediate and obvious factor affecting soil moisture levels. Temperature plays a crucial role as higher
temperatures increase the rate of evaporation from the soil and transpiration from plants, both of which reduce soil moisture
and increase vegetation flammability. Additionally, atmospheric humidity levels, which are influenced by temperature and
precipitation patterns, affect soil moisture through their impact on evaporation rates. All those effects from these weather
parameters are reflected in panels (m1-o1) and (m2-p2) as northern and central Greece appears to have the greatest decrease
in soil moisture as a response to drier weather conditions in those areas.

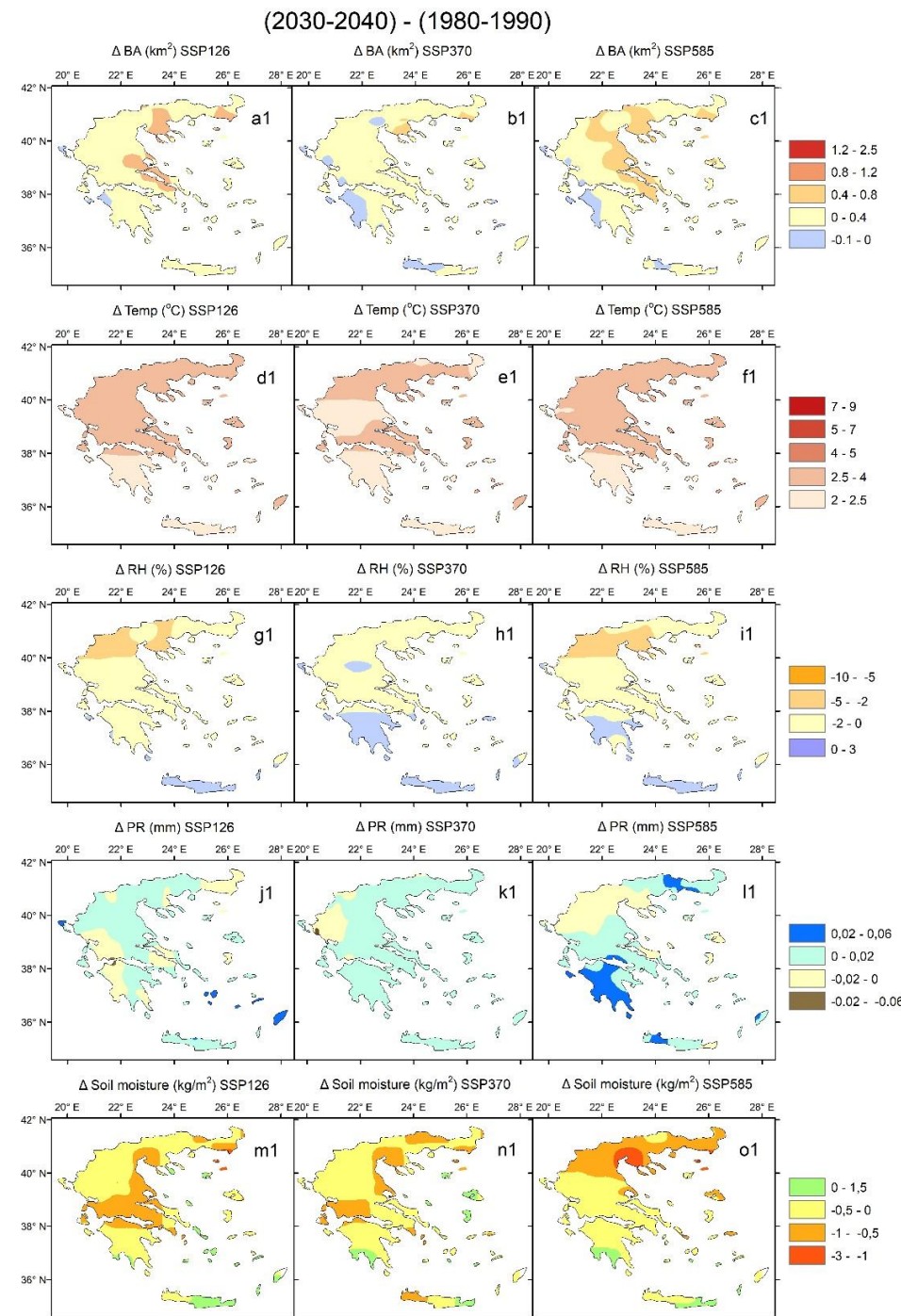

**Figure 2. Decadal mean burned area and climate variable changes for the three SSP scenarios between the near future and the reference period (panels a1-l1). Burnt area differences are shown in the 1st row. Temperature differences in 2nd row. Relative humidity differences in 3rd row. Daily precipitation differences in 4th row. Soil moisture differences in 5th row.**

(2080-2090) - (1980-1990)

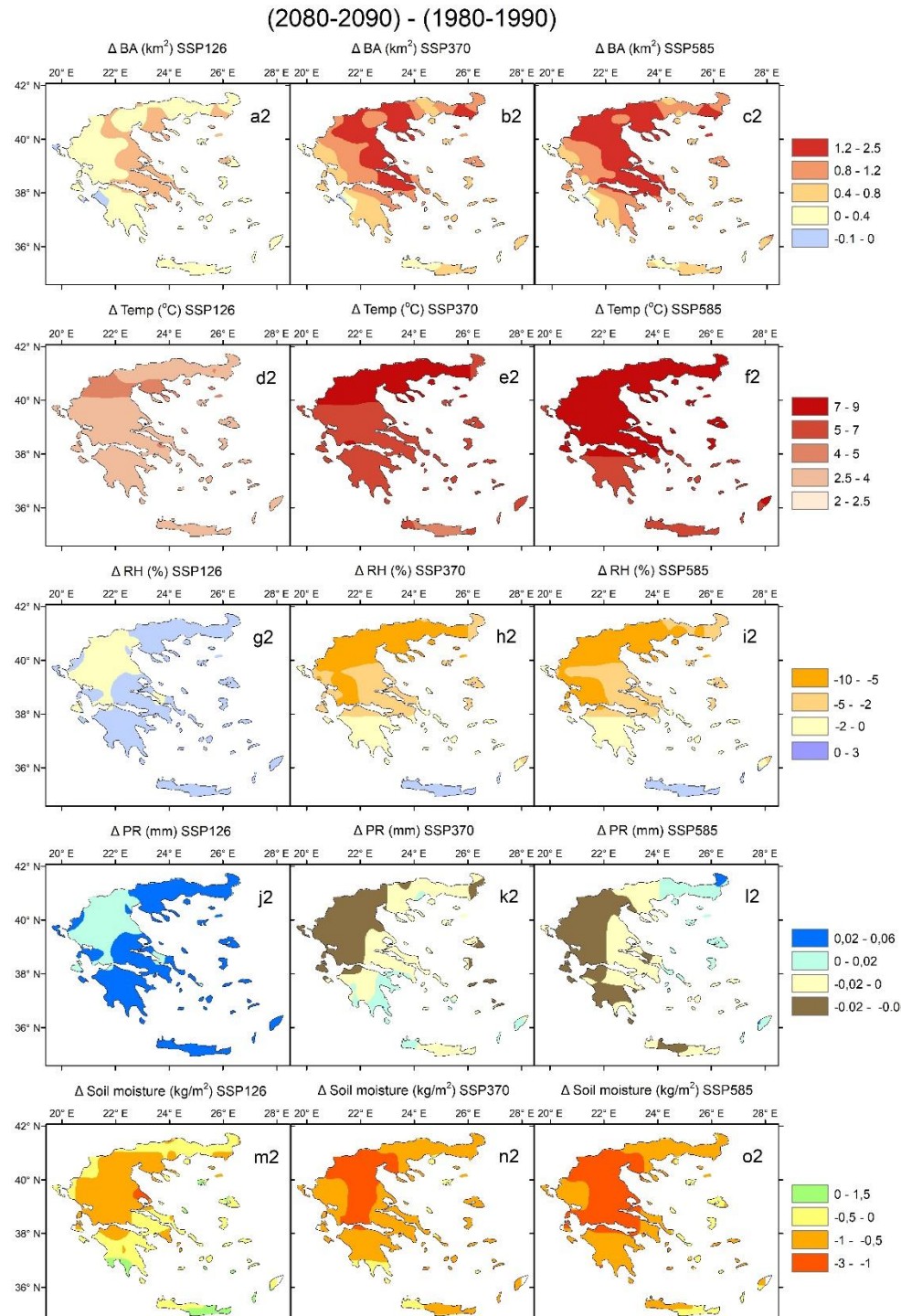

**Figure 3. Decadal mean burned area and climate variable changes for the three SSP scenarios between the near future and the reference period (panels a2-l2). Burnt area differences shown in 1st row. Temperature differences in 2nd row. Relative humidity differences in 3rd row. Daily precipitation differences in 4th row. Soil moisture differences in 5th row.**

### 3.3 Analysing fire size changes and comparison with FWI results

To better understand the changes in burnt area, we estimated changes for different sizes of burnt area from model output as described in 2.5. In Figure 4 it is evident that the majority of the burnt area changes in Greece fall in the 0-1 km² bin, across all scenarios and periods, suggesting that most areas will see little to no change in burnt area size.

Also, a substantial number of instances is represented by the small burnt area reduction bin (-1-0 km²). In that category, the number of instances is dominated by the SSP126 scenario as it suggests that climate mitigation efforts and lower emissions can lead to a reduction in the areas burnt by wildfires.

The instances with higher burnt area bins (1-2 km², 2-3 km² and 3-4 km²) follow a similar pattern in the higher emission scenarios SSP585 and SSP370 for the late century (2080-2090). Especially concerning is the presence of around 1000 additional individual instances across Greece in the timespan of 10 years (2080-2090) with burnt area of 4 km² larger than in the reference period (1980-1990). This emphasizes the potential exacerbation of wildfire activity under high greenhouse gas emissions, which could lead to more extensive and possibly more destructive wildfires.

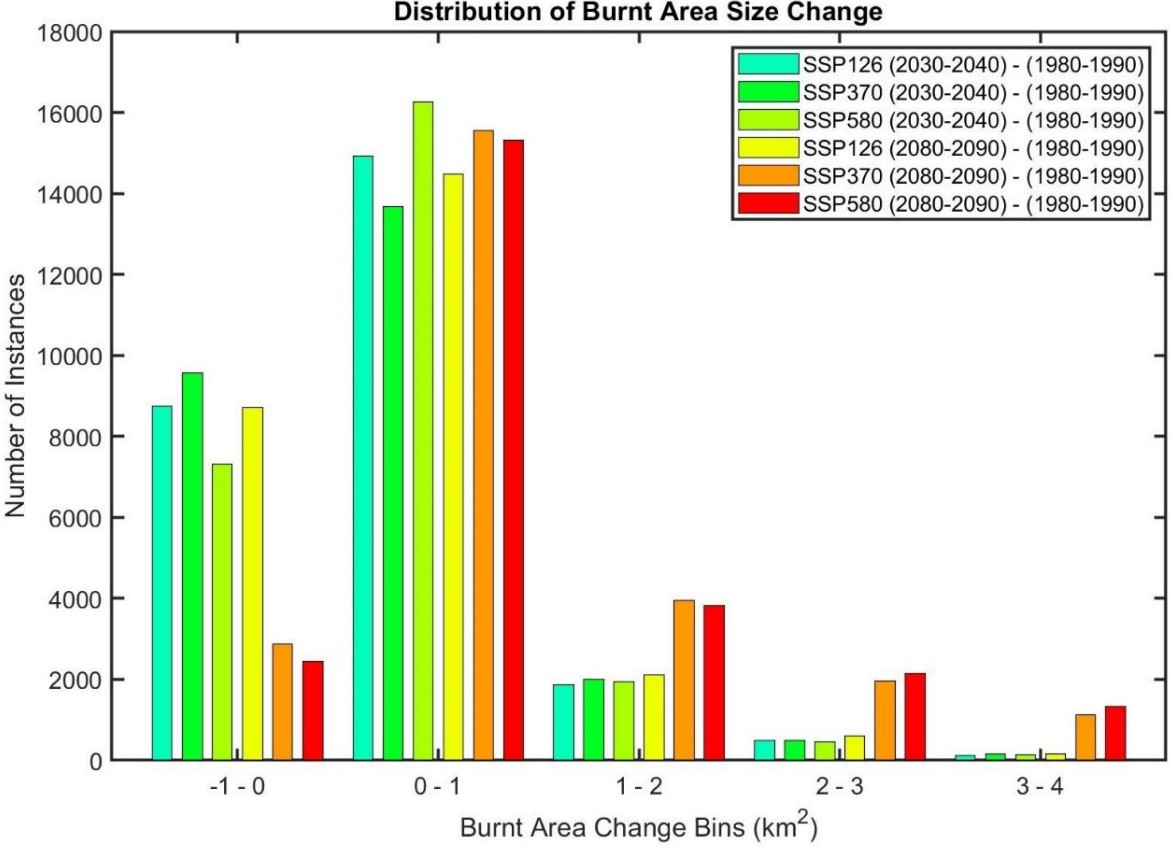

**Figure 4. Distribution of burnt area size changes under different SSP Scenarios for future periods (2030-2040 and 2080-2090) compared to the reference period (1980-1990).**

The distribution of the areas mostly affected by high burnt area events greater than 12 km² corresponding to the 99th percentile of all burnt area values during the reference period 1980-1990 can be seen in Figure 5. Panels a, b, and c show a clear pattern for the most affected areas to be in eastern continental Greece, with the most pessimistic future scenario (SSP585) showcasing the highest extent of the areas, with up to 100 additional individual events with burnt area greater than 12 km² during the two future decades 2030-2040 and 2080-2090, compared to the reference period 1980-1990.

In the monthly burnt area frequency analysis, it can be observed that the number of high burnt area events in the period 2080-2090 (panel e) practically doubles compared to the period 2030-2040 (panel d). The highest frequency of those catastrophic events occurs in July (7th month) and August (8th month) across all scenarios. It is notable to mention that even though all climatic scenarios in the period 2030-2040 do not predict any high burnt area events in May and September, this behaviour completely changes in the period 2080-2090 for all emission scenarios (except for SSP126), with September featuring 20 more high burnt area events across Greece. May, on the other hand, does not demonstrate any sizeable changes. That potential increase in catastrophic events towards the end of the fire season is in line with the findings from our previous study using the FWI instead of a fire model (Rovithakis et al. 2022), where we showed that the fire season length is expected to increase by up to a month, with a future October resembling a present-day September.

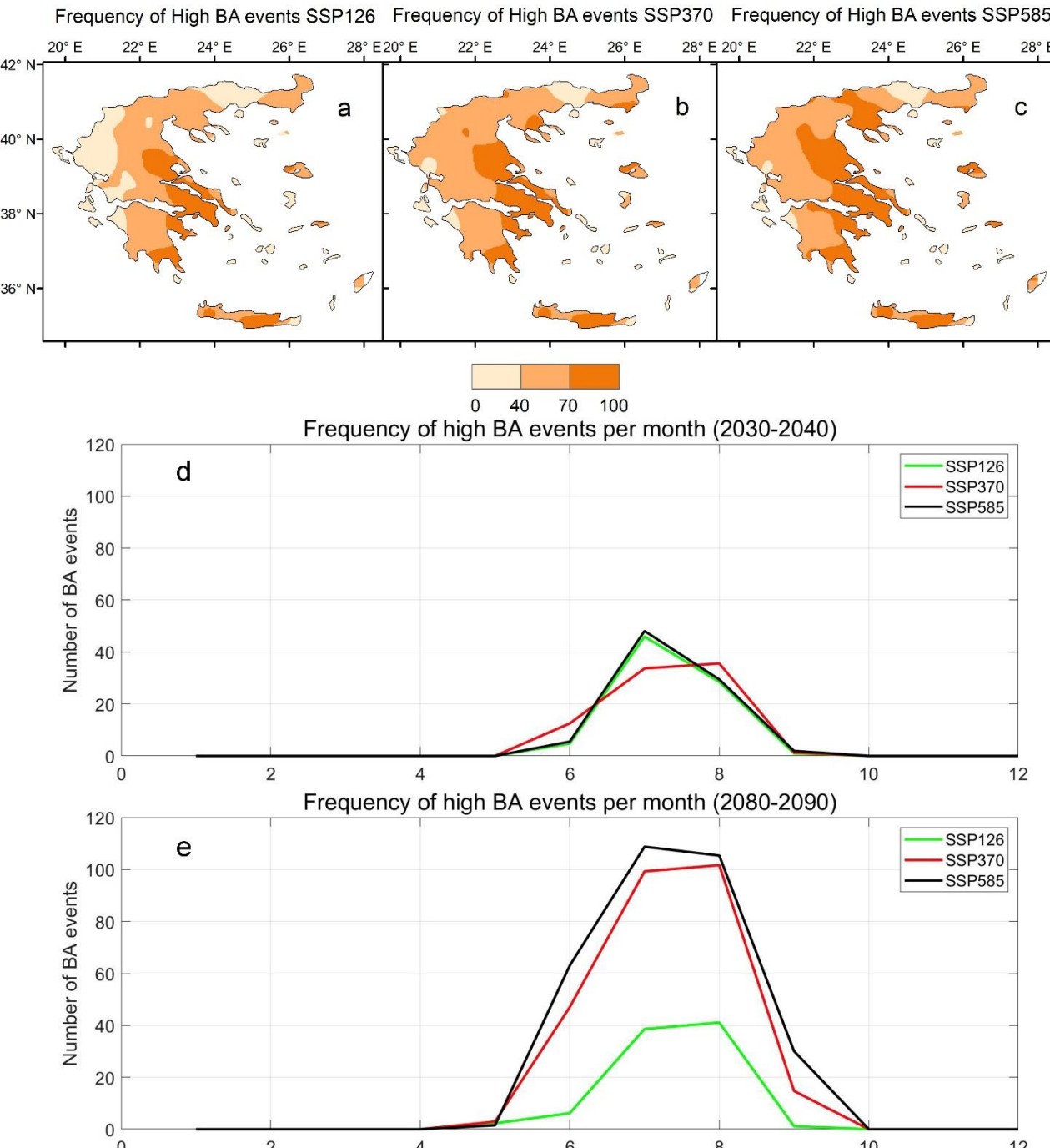

**Figure 5. Frequency of high burned area (BA) events greater than the 99th percentile of the reference period in Greece under different SSP scenarios. Panels a, b, and c show the spatial distribution of the difference of the frequency of high BA events between the periods 2030-2040 and 2080-2090 under SSP126, SSP370, and SSP585 scenarios, respectively. Panels (d) and (e) display the temporal distribution of the frequency of high BA events per month aggregated for all of Greece for the periods 2030-2040 and 2080-2090 respectively under the same SSP scenarios.**

By temporally averaging the burnt area results for all SSP scenarios for the reference (1981-1990) and the two future periods (2031-2040 and 2081-2090), we qualitatively compare them with the Canadian Fire Weather Index (FWI) results from our

previous study (Rovithakis et al. 2022) for the equivalent time periods (Figure 6). A similar pattern of changes can be seen, with the reference period showing the smallest changes, the distant future in SSP585 showing the greatest changes, and the rest of the cases showing changes with magnitudes in between. This agreement between FWI-based results and results coming from a full-on fire model demonstrates the FWI's good skill in terms of capturing future burnt area tendencies as seen in Figure 6.

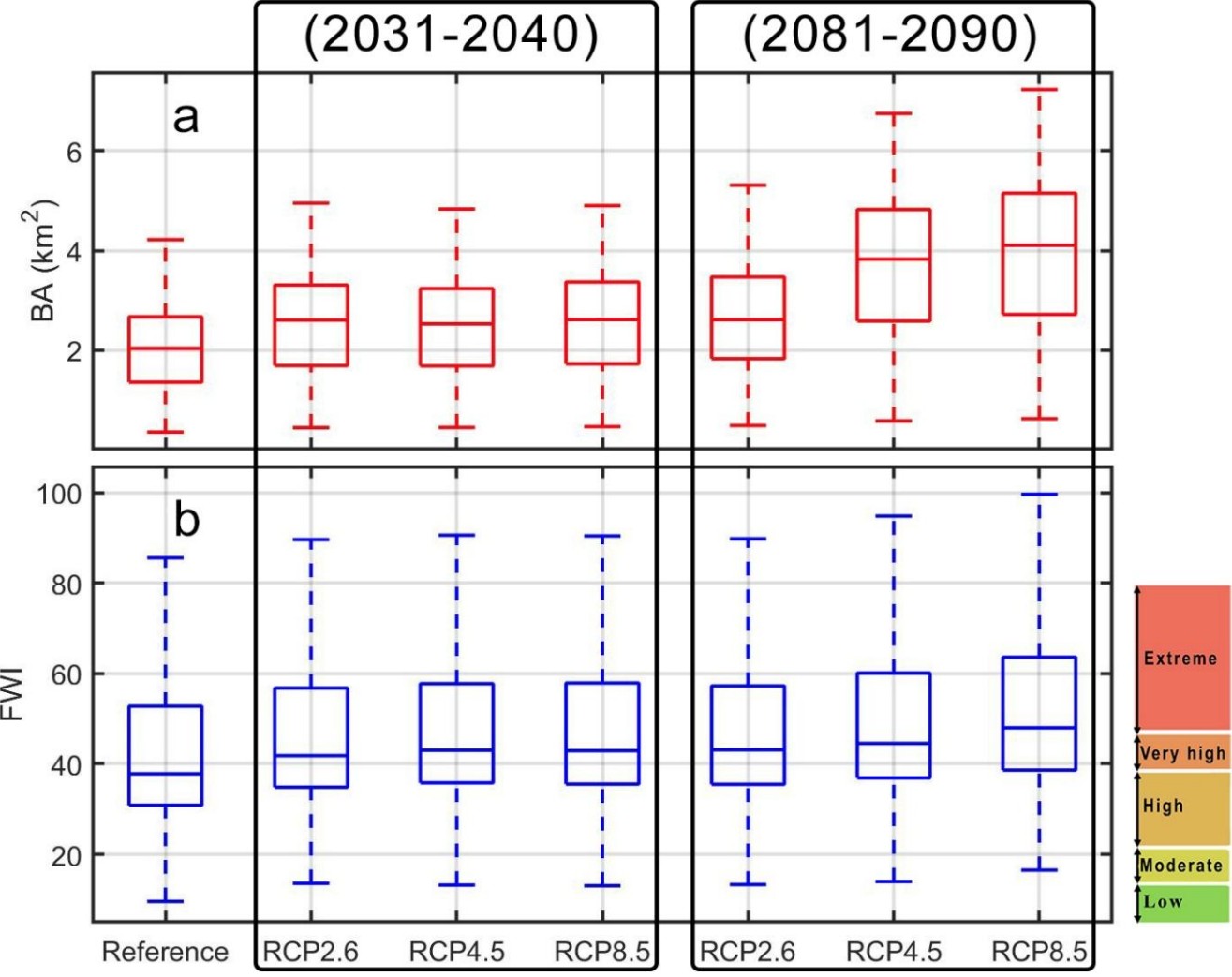

**Figure 6. Boxplots showing the evolution of burnt area for the three SSP scenarios using JULES-INFERNO (panel a), compared to the FWI for the same time periods as calculated from our previous study (Rovithakis et al. 2022) (panel b).**

### 3.4 The role of static vs dynamic vegetation

Subsequently, we compare the changes between the distant future and the near-future [(2081-2090) – (2031-2040)] in the simulations with dynamic versus static vegetation in terms of burnt area (Figure 7). We do not compare with the reference period used in previous sections (1980-1990), as the years corresponding to it were not included in the pre-existing ISIMIP3b simulation featuring dynamic vegetation (which, started from year 2015).

Comparing Figure 7 panels (a-c) with (d-f), two main features become apparent: 1) that future burnt area is decreasing in some areas in northern Greece when including dynamic instead of static vegetation, and 2) that the opposite is true for southern Greece, where burnt area is projected to increase slightly everywhere. The decreases in burnt area in northern Greece with dynamic vegetation (panels g-i) could imply a climate-induced shift towards less flammable vegetation types or changes in fuel continuity, while increases in the south suggest either the persistence or expansion of fire-prone vegetation or an exacerbation of fire conditions in already flammable landscapes which is in line with (Loepfe et al., 2010).

To explain the difference in burnt area patterns between panels (b-c) and (e-f), the carbon mass availability in different vegetation types was examined for a rectangular region representing the areas with the largest difference in panel (i) corresponding to northern Greece ('NG') and another region representing the areas with the smallest difference in panel (i) corresponding to southern Greece ('SG') (Figure 8, panels a-f).

Figure 8 panels (a-f) show the trends of biomass in different types of grass and trees. The latter demonstrate a longer periodicity than grass types since forests have a greater resilience to yearly weather fluctuations. The NG areas with the the most pronounced increase in burnt area show a decreasing tendency in evergreen needleleaf trees (black line) as seen in Figure 8 panels (a-c). On the other hand, SG areas not only show a small increasing tendency in the same type of evergreen needleleaf trees, which is a drought resistant vegetation category, but also have the evergreen broadleaf trees (red line) as the second highest carbon mass, showing an increasing tendency and explaining why the dynamic vegetation burnt area increases in SG areas in Figure 7 panels (e-f). Similar projections of evergreen needleleaf and broadleaf trees (which are flammable and adapted to heat) showing the strongest and most dominating response to severe drought were also found for other Mediterranean countries (Fang et al., 2021). This shows how climate change can favour certain vegetation types that, in turn, influence future fire susceptibility, creating important ecosystem feedbacks which was also found by Burton et al., (2019)

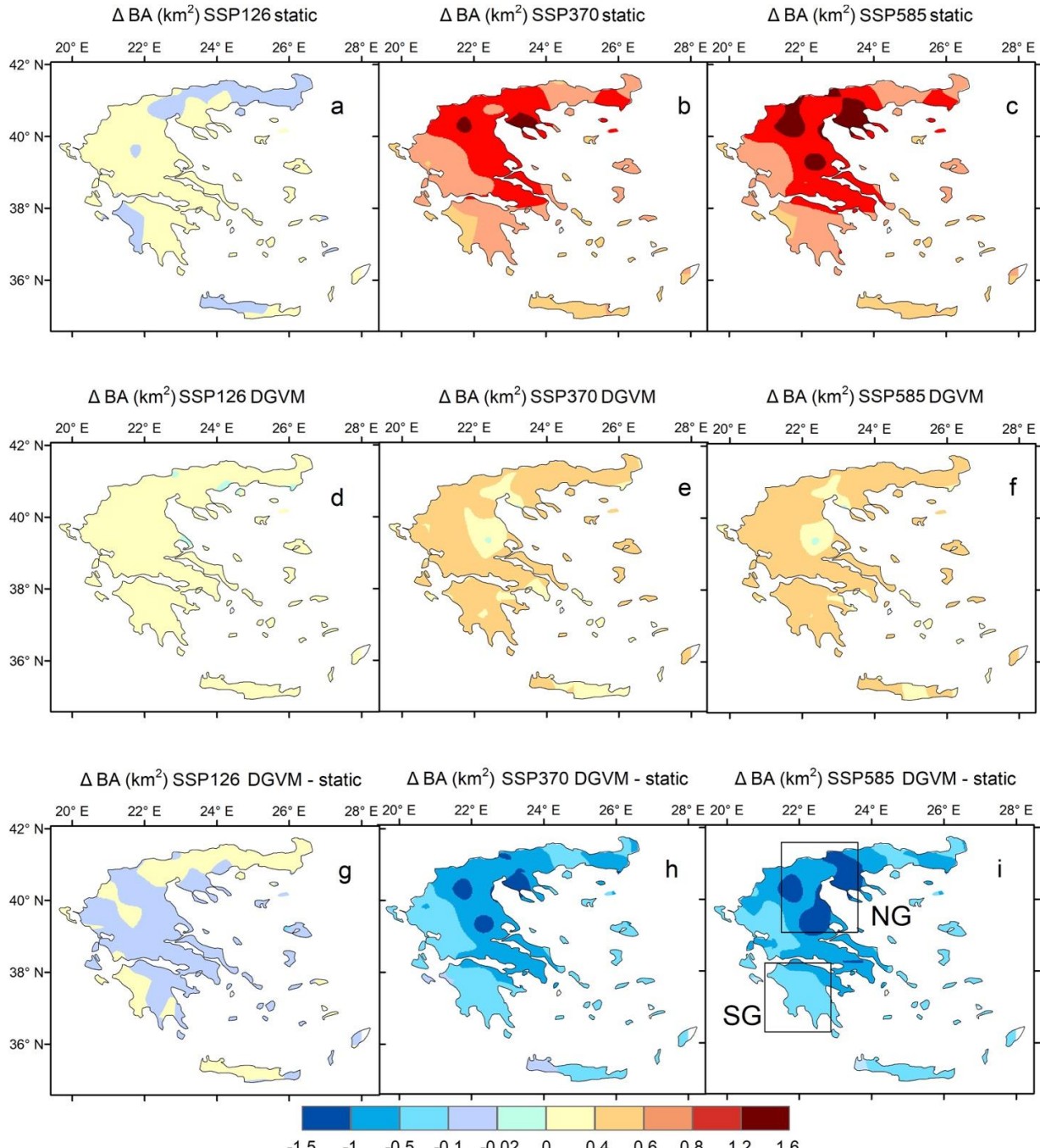

Figure 7. Burnt Area differences between distant future and near future from the simulation with static vegetation for the 3 SSP scenarios panels (a-c). Burnt area differences between distant future and near future from the simulation with dynamic vegetation for the 3 SSP scenarios panels (d-f). Panels (g-i) show burnt area differences between the simulation with static and dynamic vegetation. The rectangles in panel (i) define the two areas (NG and SG) within the domain the feature the strongest positive and

negative differences when activating dynamic vegetation.

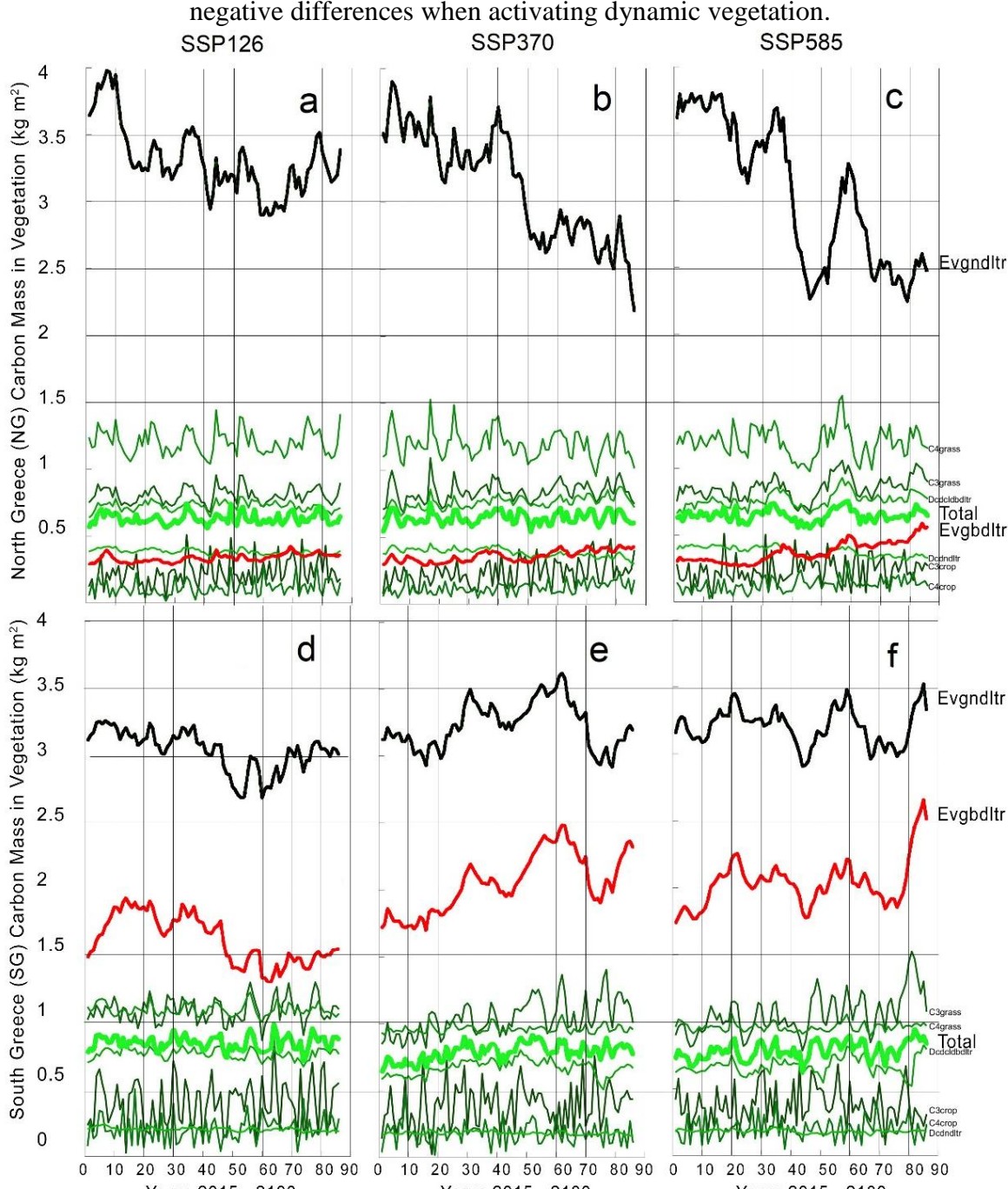

Figure 8. To explain the differences in Figure 7 for the areas NG and SG, panels (a-f) show comparisons of JULES carbon mass in different vegetation types for the years 2015-2100 taken from the ISIMIP3b simulation for each of the three SSP scenarios, averaged for northern Greece (NG, top row) and for southern Greece (SG bottom row). The vegetation types shown are: Evgndltr (evergreen needleleaf trees), Evgbdltr (evergreen broadleaf trees), C3grass (cool-climate grass, less efficient photosynthesis), C4grass (warm-climate grass, efficient photosynthesis), Dcdcldbdltr (deciduous broadleaf trees), C3crop (cooler climate crops, less efficient under high temperatures), C4crop (heat-tolerant crops), Dcdndltr (deciduous needleleaf trees) , and Total (total carbon mass in vegetation).

## 4 Conclusions

The present study evaluated future burnt area changes in Greece under future climate conditions, using a fire model (JULES-INFERNO) with static vegetation and with dynamic vegetation. The historical model-simulated burnt area with static and dynamically changing vegetation was evaluated against GFED5 observations. It was found that overall, the two simulations can capture the general characteristics of the observed burnt area, even though years with anomalously high burnt area cannot be captured in full due to the lack of real fire ignition data in such simulations.

Future climate change plays a crucial role in shaping future wildfire activity. While a substantial portion of areas may remain stable in terms of burning, there are 1000 individual wildfire events with 4 km² larger burnt area under high emission scenarios. This highlights the importance of climate mitigation efforts to reduce emissions and the associated impacts on wildfire regimes. Additionally, the areas experiencing the highest frequency of large wildfire events are distributed mainly in eastern continental Greece. Thus, these areas need even better fire management strategies and preparedness measures. Additionally, an extension of the fire season has been seen in our future results, particularly due to significant burnt area increases for the month of September.

When the simulation results with static vegetation for the two future periods as seen in Figure 7 panels b and c were spatially averaged projected increases of average burnt area of 0.8 km2 for the entire domain on average, with the highest values of up to 2.5 km2 in the eastern continental Greece. This is driven by drier climatological conditions in this region in the distant future, with temperature changes of up to 9 °C, up to 10% lower relative humidity, and slightly reduced precipitation. When burnt area is solely calculated on climatological conditions, it was found that the FWI index can reflect similar changes. On the other hand, allowing the vegetation to change dynamically led to a smaller overall distant future burnt area change of 0.3 km2 on average for the entire domain since fire is no longer igniting in areas already burnt, as well as decreases of up to 0.02 km2 for the main agricultural areas of Greece. Those burnt area decreases were due to the decreases of needleleaf trees, as this type of vegetation represents the majority of carbon mass.

Our study is subject to certain limitations. There is a wide diversity between different estimates of BA between GFED5 and MCD64A1, or FireCCI. When compared to GFED5, whilst our two model simulations manage to capture the overall trend, there is also a bias mainly due to the inevitable lack of realistic fire ignition data leading to a less accurate depiction of reality. Therefore, more studies of this kind are warranted in the future. Nevertheless, our study demonstrates the threat for increased burnt area in Greece in the future, as well as a clear potential influence of vegetation changes in shaping the future trends in burning.

## Author contribution

Anastasios Rovithakis: Formal analysis, Investigation, Methodology, Software, Visualization, Writing – original draft preparation;

Eleanor Burke and Chantelle Burton: Data curation;

Matthew Kasoar, Manolis G. Grillakis and Konstantinos D. Seiradakis: Writing – review & editing;

Apostolos Voulgarakis: Conceptualization, Funding acquisition, Methodology, Project administration, Resources, Supervision, Validation, Writing – review & editing;

## Acknowledgments

This research was funded by the Leverhulme Centre for Wildfires, Environment, and Society through the Leverhulme Trust (grant number RC-2018-023). AV has also been supported by the AXA Research Fund (project 'AXA Chair in Wildfires and
395 Climate') and by the Hellenic Foundation for Research and Innovation (Grant ID 3453). The present work was supported by the project "Support the upgrading of the operation of the National Network on Climate Change (CLIMPACT)" of the General Secretariat of Research and Technology under Grant "2023NA11900001" and by the framework of the National Recovery and Resilience Plan Greece 2.0, funded by the European Union – NextGenerationEU (Implementation body: HFRI; Project number: 015155).

## Data availability

The data to recreate the findings of this study are openly available at the following URL/DOI:

Input weather variables:

Hisotrical: https://data.isimip.org/10.48364/ISIMIP.982724.2

Future: https://data.isimip.org/search/tree/ISIMIP3b/InputData/climate/atmosphere/ukesm1-0-ll/

GFED5 Burnt Area observations:

https://doi.org/10.5281/zenodo.7668423

JULES-INFERNO ISIMIP3a burnt area data:

https://data.isimip.org/10.48364/ISIMIP.446106

The JULES code used in these experiments is available for non-commercial use on the JULES trunk from version 4.8 (revision
6925) onwards. The rose suite used for these experiments is u-cc669 at vn6.2 (located in the repository at trac/rosesu/log/a/p/8/4/5 r69824). Both the suite and the JULES code are available on the JULES FCM repository: https://code.metoffice.gov.uk/trac/jules (registration required).

## Conflict of interest

The authors declare no competing financial interests.

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
