# Peer review of "Estimating future wildfire burnt area over Greece using the JULES-INFERNO model"

_EGUsphere, 2025_

## Author Response (AR1)

**Estimating future wildfire burnt area over Greece using the JULES-INFERNO model**

Anastasios Rovithakis, Eleanor Burke, Chantelle Burton, Matthew Kasoar, Manolis G. Grillakis, Konstantinos D. Seiradakis, and Apostolos Voulgarakis

**Referee 1**

It is an intersting topic however I beilive that the manuscript needs further development for becoming accepted.

We would like to thank the reviewer for the comments provided. All the comments below are now addressed and have made our manuscript stronger.

My comments for the authors are the following:

1. They extensively use acronyms which makes the flow of the reading quite difficult.

This is a shared comment between the referees which we have now addressed.

2. In the Introduction, the way it is wriiten I think the readers need extensive prior knowledge of the topic, so that they are able to follow the text. I think the authors need to provide more explanations and text relevant to what they are doing.

We added more information in the introduction to make it easier to follow for the readers.

3. In the section 2 the authors have written everything, the data, the methods the background of their model. I think it should be split to the corresponding sections (background, data, methods) and developed more.

Indeed this section was poorly structured. We have now split this section into several ones and added considerably more information to have a more well rounded section.

4. My biggest concern is the performane of their model, as this is presented and discussed in figure 1d. I find that this model completely misses the fires occured especially in the year 2007, 2011 and also the 2019 where there are no fires. What is the correlation between the observed and estimated? The authors support that their model does not consider ignition sourses. Then, maybe they need to change even the title of their work and try to estimate maybe the conditions (meteorological) suitable for having fires.

We acknowledge in 3.1 that fire models such as JULES-INFERNO primarily rely on weather conditions and vegetation quantities to calculate burnt area. A key limitation is that

they operate "without any information on actual fire ignitions" that occurred in reality. The stochastic nature of real fire ignitions makes them "impossible to predict".

This limitation is explicitly stated as the reason why the model cannot capture the extremely high 2007 actual burnt area seen in Figure 1 panel (d). However, the model *does* capture "the tendency for increased burnt areas in years such as 2007 that featured increased fire weather". So the model aims to capture the general behaviour driven by climate and fuel, and evidently it has a decent skill in doing so.

When it comes to the JULES-INFERNO simulations when compared to the GFED5 observations, the former can still capture the general burnt area behaviour with an acceptable correlation for the majority of the areas in the Greek domain as seen in Figure 1 panels a and b, especially for the southern parts of the country. We also added information from other studies showing that JULES is validated and is able to capture burnt area trends. That is why we believe there is not reason to change the title, as it reflects well the scope of this research.

5. Also, I suggest the auhtors to evaluate spatially the estimated vs the observed fire activity. Where in space toberve th fires and compare this to their estimates from the model.

We did perform a spatial evaluation of the model's performance against observed fire activity. Specifically, the temporal correlation between the simulated burnt area (both with static and dynamically changing vegetation) and the GFED5 observations was examined and presented spatially. This is visualized in Figure 1 panels a and b, which show maps of Greece with color gradients indicating the correlation coefficient for different areas. This evaluation was performed for the years 2004 to 2019 due to the overlap of burnt area observation and modeled output for static and dynamic vegetation.

**Referee 2**

The manuscript entitled "Estimating future wildfire burnt area over Greece using the JULES-INFERNO model" aims to analyse the potential impacts of future climate change and vegetation changes on area burned over Greece. The authors applied the JULES simulator (Joint UK Land Environment Simulator) coupled with the INFERNO fire model, and used future climate projections from the UKESM1 model as input data.

The work presents a set of limitations and critical aspects that need to be addressed by the authors. In the Specific Comments, a list of points that should be verified, clarified or improved is provided

I recommend major revisions of the manuscript before publication.

We thank the reviewer for the detailed comments provided. All of them have now been addressed.

SPECIFIC COMMENTS

L25-64: The Introduction would benefit from a better review of the most relevant literature in the topics under investigation, for instance increasing the number of references to appropriate and more recent studies.

Indeed this section was lacking some detailed explanation. We have substantially revised the Introduction to incorporate a wider range of appropriate studies.

L33 (and in other parts of the manuscript): please replace "oC" with "°C". This is also valid for "o" (degrees), which should also be amended.

We thank the reviewer for noticing and have replaced these now.

L37 (and elsewhere): please amend the references.

The references have been amended.

L44-49: These sentences need to be improved and possibly should include references to more recent studies when necessary. For instance, the fact that "Landscape fragmentation is projected to increase in the Mediterranean region" is valid for some Mediterranean areas, while other areas could face opposite patterns in future years (and are facing opposite patterns in current years). In the Salis et al work, the unmanaged vegetation overgrowth is related to agricultural land abandonment rather than to fire suppression. Again, the projected agricultural expansion due to improved technology balancing the effects of climate change should be better contextualized.

We agree that this part was convoluted. We have now added and clarified this information using some additional studies.

L65-125: The methodological part needs to be better organized, as the presence of a single Section ("Data and JULES model setup") is not satisfactory. The methods did not describe the approach used to obtain data and findings presented in the Results Section, or how wildfire size was derived. The use of FWI is not mentioned in this section. Considering that

the authors are emphasizing the potential effects of future conditions on wildfire size in Greece, describing how future wildfire size was estimated is crucial. In addition, I would recommend including a figure that summarizes the methodological approach adopted.

The methods section was indeed unstructured. We added several subsections making it easier to follow, as well as new information about how the wildfire size was derived and the use of the FWI. With all these detailed changes we now believe that we have encapsulated the adopted methodological approach. We would argue that a figure is not necessary, as the methodological approach is not particularly convoluted. A schematic of the INFERNO model's approach can be found in the manuscript of Mangeon et al. (2016) describing the model.

L69-71: Wind speed is a key driver for the occurrence of large wildfires and strongly influences wildfire regime. If the model does not consider this input, this is a significant limitation that should be mentioned in the paper.

Wind speed can indeed be a key driver for large wildfires and its absence in the model is a recognisable limitation. It is indeed important when conducting high resolution simulations to determine the effects and propagation of individual wildfire events. However, for large-scale simulations we are interested in the collective effect of a range of fires, without focusing on the specific spread of individual fires. It was found during the INFERNO model developments that introducing wind speed as a factor in the model would not increase the skill of capturing burnt area on the spatial and temporal scales for which the model is typically used. We now clarify that in the manuscript.

L74: Please clarify if the fuel load index only refers to surface fuels, or if canopy fuels are also included. Moreover, how were the biomass data estimated (sources of data?)?

Both are included and the biomass data are estimated internally by JULES. We now mention that.

L87-88: The plant functional types (PFTs) should be better described. More specifically, please clarify how many categories were used for the study area, and provide a PFT map.

We now added much more information about the PFTs. The categories themselves are specified in section 2.2. When it comes to the PFT map the reason why we did not include it is because of the data structure which represents all PFTs as percentages for every grid. So we cannot plot one map showcasing all PFTs.

L91-92: Do you mean that the number of simulated fire ignitions is constant for current and future scenarios? This contrasts with previous works that indicate a potential lengthening of the fire season (see L31-34) and an inherent increase in future fire occurrences. Please clarify.

We kept the number of simulated fire ignitions constant for current and future scenarios in order to determine how the increase in fire occurrences, burnt area, and fire season length are driven specifically by the climate-induced changes in vegetation flammability and vegetation dynamics. We now clarify this better.

We also note that fire ignitions and fire occurrences are two different quantities, with occurrences being the result of flammability and ignitions combined.

L96: "Our simulation domain covers the entire globe". Is this correct?

Yes, our initial simulations were global and then were cropped over the Greek domain to keep consistent boundary conditions. We mention that now.

L96-113: there are many acronyms, and this does not help reading this part of the paper.

This section is indeed hard to read without prior knowledge of the acronyms so now we explain them better.

L132: "from the obsclim climate experiment". Please be more precise.

We now explained this type of climate experiment based on observation reanalysis and clarified the acronyms.

L147: "increased fire weather". This should be amended

The reviewer is correct this is information we know from our previous study but it does not have a place here. We amended it.

L150-154: please define, in both the legend and the caption, the unit of measure of Figures a, b, and c. Please also clarify if you are referring to decadal, annual, or monthly area burned values.

Panels a and b in figure 1 are temporal correlations with panel c being their difference so they are unitless. That is why we did not define them. The burnt area values in panel d are annual averages - we now mention that.

L162-167: Again, it is not clear if the area burned refer to decadal or annual values.

We thank the reviewer for noticing. These are decadal means based on daily values.

L189-191 & L193-195: The units of measure should be checked and clarified when necessary. The temporal scale of the units (e.g.: monthly, decadal, annual averages?) should be defined.

As mentioned above we now added this important information.

L198: "To better understand the changes in burnt area, we estimated changes for different sizes of fires". How? Using the model or FWI outputs? This should be defined in the Methods.

For this, we used modeled burnt area output, not the FWI. We now added in Sect. 2 a detailed description of how this figure was made.

L271-275: The use of different climate change scenarios in the work needs to be justified. In more detail, the authors proposed RCP2.6, RCP7.0 and RCP8.5 for the analysis of static vs. dynamic vegetation models. The FWI analysis considered RCP2.6, RCP4.5, and RCP8.5. The wildfire size analysis was based on SSP126, SSP370, and SSP585 scenarios. Why? This is a key point that needs to be addressed, as the climate data are different but combined in the Results without any critical consideration.

We thank the reviewer for noticing. Figure 7 was not updated but all these panels were made using SSP126, SSP370, and SSP585 as can be seen in the caption. We now updated the figure to be correct. As for the FWI analysis and specifically panel b, this was a figure taken from our previous published paper. As we mention in the manuscript, we added the figure to *qualitatively* compare burnt area results for all SSP scenarios with the FWI. Even though we are comparing different types of future scenarios, this figure is still important to compare how the FWI trend compares qualitatively with the burnt area trend, since these SSPs are coupled with Representative Concentration Pathways (RCPs) to define the overall climate forcing. The Fire Weather Index (FWI) data is directly derived from climate model outputs forced by these RCPs (RCP2.6, RCP4.5, RCP8.5). Therefore, this comparison is valid as it examines the relationship between a direct climate-driven fire danger metric (FWI) and a fire impact metric (BA) under consistent underlying climate change projections, allowing to assess the influence of similar climatic shifts on both fire weather and resultant burnt area. We now mention that in the manuscript.

Results: the work is missing the discussion of the results and findings in the light of previous similar studies, as well as the discussion of the significance of the different results presented. I recommend including a "Discussion" section, or improving to Results by discussing the results in relation to those of others.

We agree that a more thorough discussion of our results, in the context of existing literature would substantially improve the manuscript. We have integrated it directly into the "3. Results" section, following the presentation of the figures. We believe this approach allows for a more cohesive narrative, making it easier for the reader to follow the implications of the findings.